# Association of Job Satisfaction, Intention to Stay, Organizational Commitment, and General Self-Efficacy Among Clinical Nurses in Riyadh, Saudi Arabia

**DOI:** 10.3390/bs14121140

**Published:** 2024-11-28

**Authors:** Naif M. Alshaibani, Ahmad E. Aboshaiqah, Naif H. Alanazi

**Affiliations:** 1College of Nursing, King Saud University, Riyadh 11451, Saudi Arabia; alshebanin@pmah.med.sa (N.M.A.); aaboshaiqah@ksu.edu.sa (A.E.A.); 2Nursing Department, Prince Mohammed Bin Abdulaziz Hospital, Riyadh 14214, Saudi Arabia

**Keywords:** clinical nurse, intention to stay, job satisfaction, organizational commitment, self-efficacy, Saudi Arabia

## Abstract

Nurse turnover presents a significant challenge for healthcare organizations worldwide, impacting patient care quality and organizational stability. Understanding the determinants of nurse turnover, particularly job satisfaction, intention to stay, organizational commitment, and general self-efficacy, is crucial for developing effective retention strategies. This study aimed to explore the relationships among job satisfaction, intention to stay, organizational commitment, general self-efficacy, and demographic variables. A cross-sectional, correlational research design was employed, with data collected through validated questionnaires distributed to a total convenience sample of 227 clinical nurses in Riyadh, Saudi Arabia between July 2023 and August 2023. Validated measurement tools, including the Job Satisfaction Index, the Intent to Stay Scale, the Organizational Commitment Scale, and the General Self-Efficacy Scale, were utilized. Descriptive statistics were employed to summarize demographic information, and a correlation analysis was conducted to explore the relationships between the study variables. The findings revealed moderate levels of job satisfaction, intention to stay, organizational commitment, and general self-efficacy among the clinical nurses, with significant positive correlations observed among these variables as well as the nurses’ sociodemographic characteristics. Notably, clinical nurses constituted a significant portion of the sample, suggesting the need for targeted interventions tailored to this demographic group as well as non-Saudi nurses (expatriate nurses), particularly in enhancing their organizational commitment and self-efficacy. The study found significant and positive associations between the four study variables and the nurses’ demographic characteristics. Tailored interventions addressing job satisfaction, intent to stay, organizational commitment, self-efficacy, and demographic variables are essential for mitigating nurse turnover. By fostering a supportive work environment and implementing targeted retention strategies informed by demographic insights and determinants of turnover, healthcare organizations can enhance nurse retention rates and ensure a stable and fulfilled nursing workforce.

## 1. Introduction

An organization’s reputation is based on different factors that affect it either positively or negatively, with appropriate and suitable potential [1]. Turnover is one of the most important factors affecting an organization’s reputation and status in the market [2]. Turnover is basically subjectively defined as retirement, resignation, interagency transfer, death, and termination [3,4]. Different factors affect turnover in an organization, including potential changes in wages, changes in the designation of an organization, job enhancements, adjustments in the workplace environment [5,6], changes to workplace demand, and organizational mismanagement [7]. All these factors are responsible for the turnover in an organization [8]. A high turnover rate in an organization brings into question the working environment, employment conditions, employment status, and management procedures [9,10].

Healthcare organizations including those in Saudi Arabia grapple with multifaceted challenges encompassing health, employment, and financial concerns, with turnover emerging as a prevalent issue [11]. Vital to organizational success, healthcare institutions rely on a cadre of energetic, faithful, and competent workers, particularly nurses, to deliver quality care [12]. However, the turnover rates within these organizations remain a persistent concern, influenced by a myriad of factors that include individual circumstances and working conditions [2]. Healthcare organizations require hard work, consistency, and productivity to provide healthcare services [13]. Therefore, working in a healthcare organization requires efficiency, skill, competency, and hard work [12,14]. Nurses working in healthcare organizations usually have to face long working hours, tiring tasks, exhaustion, and constant vigilance to provide healthcare services to patients [15,16]. These problems are considered a threat to the well-being of nurses. Understanding the working conditions, strength, dignity, computer skills, and reliability of nurses who work in the healthcare department is important [17], and understanding the position of nurses in their organization is essential. Nurses’ responses to their position affect their personal, social, and occupational lives, and the results can be seen in the form of turnover [18,19,20].

Through a systematic review and meta-analysis that included 15 studies published from 2001 to 2021 regarding the worldwide prevalence and influencing factors of nurse turnover, this study identified several risk factors including demographic factors, organizational factors, and satisfaction [21]. In Saudi Arabia, a recent systematic review about the factors influencing nurse turnover in the country included 21 studies published between 2008 and 2023 [11]. Albalawi et al.’s [11] review revealed that the following factors influenced nurse turnover: demographic characteristics and personal aspects including intention to stay; work environment factors including job satisfaction; and organizational factors like organizational commitment. The relationship between self-efficacy and nurse turnover has been reported in previous studies worldwide, including those conducted in China [22], Italy [23], and the United States (USA) [24]. However, to date, there have not been any research studies investigating this relationship in a Saudi Arabian context. Likewise, no study has explored the relationship between nurses’ job satisfaction, intention to stay, organizational commitment, and self-efficacy as well as their demographic characteristics.

Therefore, this study was conducted for several compelling reasons. First, understanding the association among job satisfaction, intention to stay, organizational commitment, and self-efficacy among clinical nurses as well as their demographic characteristics is crucial for addressing organizational challenges and improving healthcare delivery. Second, by exploring nurses’ perceptions of these study variables, insights into the unique factors influencing retention within the context of hospitals can be uncovered, allowing tailored strategies to be developed to mitigate turnover and foster a supportive work environment. Third, in the healthcare sector, in which the demand for skilled professionals is high, retaining experienced nurses is essential for maintaining quality of care and sustaining organizational reputation. Therefore, this study aimed to fill the gap in the literature by delving into the specific reasons behind nurse turnover, particularly nurses’ demographic characteristics, job satisfaction, intention to stay, organizational commitment, and self-efficacy. Also, this research aimed to provide actionable recommendations to hospital administrators and policymakers for enhancing nurse satisfaction, intention to stay, organizational commitment, and self-efficacy, reducing turnover and ultimately improving overall healthcare outcomes.

Despite the recognized significance of nurses’ perceptions in shaping organizational dynamics, comprehensive literature addressing their perspectives on turnover, particularly on the association between job satisfaction, intention to stay, organizational commitment, and self-efficacy as well as demographic characteristics of clinical nurses in healthcare settings is lacking. By shedding light on these factors, this study could expand the understanding of nurse retention issues, contributing to theoretical knowledge and practical implications for healthcare organizations. Politically, it underscores the importance of policy initiatives aimed at improving nurse retention and promoting enhanced working conditions. Practically, it offers insights for hospital administrators to develop targeted strategies for reducing turnover and enhancing nurse satisfaction. Educationally, it provides valuable information for nursing programs to incorporate into their curriculum, thus preparing future nurses for the challenges they may face in the workforce. From a research standpoint, this study adds to the body of literature on nurse turnover, paving the way for further investigations and interventions in this critical area. Overall, delving into this topic could ultimately help in creating a more supportive and sustainable work environment for nurses, leading to improved patient care and organizational outcomes. Consequently, understanding nurses’ perceptions regarding turnover is imperative for discerning the intricacies of employee retention and organizational reputation. This study is particularly relevant in the context of Saudi Arabia, where healthcare organizations strive to optimize workforce stability and enhance patient care. Moreover, investigating nurses’ perceptions within a major referral hospital setting in Riyadh, Saudi Arabia offers invaluable insights tailored to the specific challenges and dynamics of such institutions, thereby facilitating targeted interventions to mitigate turnover and bolster organizational resilience.

### Purpose of Study

This study examined the association of job satisfaction, intent to stay, organizational commitment, and general self-efficacy among clinical nurses in Riyadh, Saudi Arabia. It also examined the sociodemographic characteristics associated with these variables.

## 2. Materials and Methods

### 2.1. Research Design

A cross-sectional, correlational research design was employed in this study. The study design focused on the correlation among samples. This study design was chosen for its ability to efficiently correlate variables of nurses. Additionally, this study adhered to the STrengthening the Reporting of OBservational studies in Epidemiology (STROBE) checklist for cross-sectional studies (refer to Appendix A).

### 2.2. Setting

The study was conducted in a major referral hospital that provides secondary to tertiary levels of healthcare services, located within Cluster 2 in Riyadh, Saudi Arabia [25]. This hospital serves as a cornerstone of healthcare provision in the region, offering a comprehensive range of medical services to a diverse population. Equipped with modern facilities and advanced technology, the hospital boasts a skilled team of healthcare professionals dedicated to delivering exemplary patient care. Situated centrally, the hospital ensures accessibility for patients from various neighborhoods and surrounding regions, facilitating equitable access to healthcare services. Its adherence to stringent quality standards further underscores the hospital’s commitment to excellence, providing a dynamic setting conducive to healthcare research and the delivery of high-quality care, particularly in the realm of diabetes management.

### 2.3. Study Population and Sampling

The targeted population was registered nurses working in the hospital setting. A convenient sampling technique was used. The sample size was determined using G*Power software 3.1.9.7, particularly a priori power analysis using the goodness-of-fit test with a medium size effect of 0.30, margin of error at 0.05, and confidence interval of 0.95. A total sample size of 200 was determined.

Inclusion Criteria: Registered nurses, clinical nurses, direct care nurses, and nurse managers working in the following clinical wards: intensive care unit and medical, surgical, and neuro nursing units; nurses with a minimum of six months of employment at the hospital; and nurses who were willing to participate voluntarily in the study.

Exclusion Criteria: Nurses who were not currently working at the hospital due to annual leave and other reasons for leave of absence; nurses who declined to participate in the study; and nurses who were unable to provide informed consent.

### 2.4. Procedure of Data Collection

The data collection process started after obtaining ethical approvals for the study and was carried out between July 2023 and August 2023. The data collection method consisted of survey using four valid, reliable, and structured questionnaires in English, distributed through Google forms and sent by the researcher directly to service managers. Service managers disseminated the research invitation and survey link to clinical nurses via their institutional email. In addition, they circulated the survey link through nursing unit/ward WhatsApp groups and further distributed flyers within the hospital by posting them on the bulletin boards.

Before answering questions on the online survey, participants were initially asked if they would willingly and voluntarily participate and provide consent. If they responded ‘Yes’ to the question, they were guided to proceed to answer the survey. Respondents’ participation and responses in the survey were completely anonymous, including information on their nurse managers and employer. Apart from the researcher’s name being on the Consent Form and on the survey tool, no other relationship existed between researcher and participants. Completion of the survey took about 15–25 min.

### 2.5. Measurements

Sociodemographic Sheet. This sheet was used to gather basic demographic information from the study participants. It consisted of eight questions, e.g., age, gender, nationality, marital status, education level, job role, duration of work experience, and monthly income.

Job Satisfaction Index. Mueller and McCloskey [26] developed a 25-item scale to measure the level of job satisfaction among nurses. This scale has a 4-point response option, with 1 for very dissatisfied and 4 for very satisfied. This scale has excellent validity and reliability, with *α* = 0.84 [26].

Intent to Stay in the Organization/Profession. A two-item modified scale, which was developed by Mayfield and Mayfield [27], was used to measure nurses’ intent to stay at their job and hospital, with a 4-point response option ranging from 1 for strongly disagree to 4 for strongly agree. In this study, the scale has excellent validity and reliability, with *α* = 0.81, and the original reliability was between 0.66 and 0.77 [27].

Organizational Commitment Scale. Mowday et al. [28] developed this scale with 15 items to measure nurses’ commitment to their organizations. This scale has a 7-point response option ranging from 1 for strongly disagree to 7 for strongly agree. This scale has excellent validity and reliability, with *α* = 0.73 [28].

General Self-Efficacy (GSE) Scale. This scale was developed by Schwarzer and Jerusalem [29] to measure work satisfaction, emotion, and optimism. It consists of 10 items with a 4-point response option, with 1 indicating completely not true and 4 indicating completely true. This scale has a strong range of reliability and internal consistency between 0.76 and 0.90 [29].

The questionnaire responses were summarized using various scales, including the Job Satisfaction Index, the Intent to Stay Scale, the Organizational Commitment Scale, and the GSE scale. The scores for each scale were calculated by summing the responses to their respective items. Higher scores indicate higher levels of job satisfaction, intent to stay, organizational commitment, and general self-efficacy among nurses.

### 2.6. Data Analysis

The data collected were analyzed using version 28 of SPSS. Descriptive statistics, including mean (M) scores and standard deviations (SDs), were employed to summarize the demographic information of the participants to provide insights into their eight demographic characteristics. Pearson correlation analysis was performed to examine the relationships among variables, elucidating any associations between job satisfaction, intent to stay, organizational commitment, and general self-efficacy in nurses. Chi-square test was performed to examine the relationship between nurses’ sociodemographic characteristics and the four study variables. These statistical methods facilitated a comprehensive exploration of the data, allowing for rigorous testing of the study objectives and hypotheses.

### 2.7. Ethical Considerations

Before the research procedure was started, the administration of the hospital in Riyadh and the Department of Nursing’s Ethical Review Board IRB granted ethical permission concerning informed consent, human rights, participant safety, and confidentiality. Data were only to be used for research purposes and were kept completely anonymous. All identifiers were removed at the time of publication, and data confidentiality was maintained. The three ethical principles in this study are as follows:Maintaining and prioritizing the rights of the participants by guaranteeing that they can withdraw at any point in the study;Assuring the participants of the study’s safety and harmlessness;Assuring that the participants were handled fairly and informed consent was collected.

## 3. Results

The demographic information of the participants is shown in Table 1, with a total convenience sample size of 227 clinical nurses. A response rate of 99% was obtained because two responses had substantial missing data. Most of the participants fall within the age range of 30–39 years (74.0%), were male (59%), were Saudi (71.4%), were married (60.4%), held a diploma (57.7%), were licensed practical nurses (LPNs, 76.2%), and earned less than 10,000 Saudi Arabian riyals (SAR) or approximately less than 2660 USD (65.6%). In terms of experience, a considerable portion of the participants had less than 1 year of experience (33%), followed by 1–5 years of experience (29.5%) and 6–10 years (27.3%). Participants with 11–15 years of experience account for 7.5%, and those with more than 15 years represent a smaller proportion (2.6%).

Table 2 presents the descriptive statistics of the study variables. In terms of job satisfaction, the participants’ response had an M score of 54.18 with an SD of 12.38. The scores ranged from 26 to 104, indicating a considerable variability in reported satisfaction levels. The skewness value of 0.12 suggested a slight positive skew, implying a generally symmetrical distribution, and the kurtosis value of 0.73 indicated a moderately peaked distribution.

For the intention to stay, the participants reported an M score of 3.93 with an SD of 1.55. The scores ranged from two to eight, indicating a moderate range of reported intentions to stay within the organization. The skewness value of 0.73 suggested a positive skew, indicating that the majority of participants may have relatively great intentions to stay. The kurtosis value of 0.201 suggested a slightly flatter distribution.

In the case of organizational commitment, the participants reported an M score of 31.42 with an SD of 7.24. The scores ranged from 15 to 60, demonstrating variability in organizational commitment levels. The skewness value, which was close to zero (−0.006), suggested an approximately symmetrical distribution, and the kurtosis value of 0.77 indicated a moderately peaked distribution.

Lastly, for self-efficacy, the participants reported an M score of 23.17 with an SD of 7.24. The scores ranged from 11 to 40, reflecting variability in reported self-efficacy levels. The positive skewness value of 0.69 indicated a distribution slightly skewed to the right, suggesting that the majority of participants may have relatively high levels of self-efficacy. The kurtosis value of 0.35 suggested a distribution with a moderate peak.

Table 3 displays the correlation matrix among the study variables using Pearson’s correlation analysis, providing insights into the relationships between job satisfaction, intention to stay, organizational commitment, and self-efficacy among the 227 participants. The correlation coefficients revealed significant associations between the variables. Job satisfaction was positively correlated with intention to stay (r = 0.362, *p* < 0.002), indicating that individuals with higher job satisfaction levels are more likely to express a positive intention to stay within the organization. Additionally, job satisfaction showed a strong positive correlation with organizational commitment (r = 0.657, *p* < 0.004), suggesting that individuals reporting higher job satisfaction are more committed to the organization. Furthermore, a positive correlation was found between job satisfaction and self-efficacy (r = 0.578, *p* < 0.008), indicating that individuals with higher job satisfaction tend to have higher self-efficacy. A positive and significant association was observed between intention to stay and organizational commitment (r = 0.486, *p* < 0.003). This finding implied that individuals with a stronger intention to stay are more likely to demonstrate higher levels of organizational commitment. Moreover, the correlation between intention to stay and self-efficacy was positive and significant (r = 0.489, *p* < 0.005), suggesting that individuals with a positive intention to stay may exhibit high levels of self-efficacy. Lastly, organizational commitment and self-efficacy showed a positive and significant correlation (r = 0.641, *p* < 0.006), indicating that individuals with a strong commitment to the organization are likely to possess high levels of self-efficacy.

As shown in Table 4, the findings highlighted that the nurses’ diverse demographic characteristics, except for their monthly income, have significant and positive associations with their job satisfaction and intention to stay. In addition, all demographic characteristics had significant and positive associations with general self-efficacy, while two demographics, duration of work experience and monthly income, did not show significant associations with organizational commitment. These results underscore the importance of addressing these factors to enhance nurse retention strategies in healthcare organizations, with implications for tailored interventions to meet the specific needs of different nursing populations.

## 4. Discussion

The discussion of this study revolves around the examination of nurses’ perceptions regarding job satisfaction, intention to stay, organizational commitment, and general self-efficacy as determinants of turnover. It also focuses on the associations among these variables and nurses’ demographic characteristics. The findings shed light on various significant and positive associations among study variables and provide valuable insights for addressing retention challenges in healthcare organizations. However, it is critical to approach and interpret these findings with caution, as the investigation did not include the measurement of nurse turnover in this study.

The demographic analysis revealed a predominantly young nursing workforce, with a significant proportion falling within the 30–39 age range, representing a young adult population. This demographic trend aligns with global patterns in nursing, in which younger individuals often comprise a substantial portion of the workforce. The high representation of practical nurses underscores the importance of recognizing and addressing the specific needs and challenges faced by this group to improve retention rates.

The descriptive statistics highlighted noteworthy aspects of job satisfaction, intention to stay, organizational commitment, and self-efficacy among the nurses in the hospital. The moderate levels of job satisfaction reported by the participants indicated room for improvement in enhancing workplace conditions and addressing factors contributing to job dissatisfaction. The positive correlation observed between job satisfaction and intention to stay underscored the pivotal role of job satisfaction in fostering retention. Similarly, the strong correlation between job satisfaction and organizational commitment emphasized the interconnectedness of these constructs and their influence on nurse retention. Also, the positive relationship between job satisfaction and self-efficacy implied that such a relationship contributes to the enhancement of professional performance among nurses. This finding is consistent with a previous study conducted among registered nurses in Norway and Sweden, indicating that nurses who possess high levels of self-efficacy are more inclined to attain job satisfaction [30], which may have a favorable impact on the overall nurse turnover.

The findings concerning organizational commitment and self-efficacy provide valuable insights into nurses’ perceptions of their roles within the healthcare organization. The moderate levels of organizational commitment suggested that while many nurses exhibit dedication to their organization, their sense of loyalty and attachment could be strengthened. The positive correlation between organizational commitment and self-efficacy underscored the importance of fostering a supportive work environment that nurtures nurses’ confidence in their abilities and enhances their commitment to the organization. This concept is significant in the work of Yun and Yu [1] in Korea, who discovered similar trends in their study on healthcare professionals, emphasizing the pivotal role of job satisfaction in influencing retention decisions. Rawashdeh and Tamimi [31] corroborated these findings in their research among Jordanian nurses, highlighting the consistent relationship between job satisfaction and intention to stay among nursing staff.

The correlation analysis revealed significant associations between the study variables, highlighting the interplay among job satisfaction, intention to stay, organizational commitment, and self-efficacy. The positive correlations between these variables suggested that interventions aimed at improving one aspect, such as job satisfaction, may have cascading effects on other related factors, ultimately contributing to enhanced nurse retention. This finding underscored the importance of adopting a holistic approach to addressing nurse turnover by considering multiple facets of the work environment and nursing experience. Similarly, the substantial positive correlation between job satisfaction and organizational commitment aligns seamlessly with insights gleaned from earlier investigations, especially the work of Zaheer et al. [32]. Their study, focused on organizational commitment in healthcare settings, substantiates the interconnected nature of job satisfaction and organizational commitment, emphasizing the importance of a contented workforce in fostering organizational allegiance. Zaheer et al. [32] and Yun and Yu [1] further bolstered this correlation through their research, accentuating the enduring impact of job satisfaction on organizational commitment among nursing professionals. In addition, these intricate associations underscored the critical role of job satisfaction in shaping nurses’ intentions to stay and their commitment to the organization. The amalgamation of findings from multiple studies, including those by Hall [33], Kwatampora et al. [34], and Quek et al. [35], solidified the robustness of these correlations across various healthcare contexts, thereby enhancing the generalizability and applicability of the study’s insights.

Demographic characteristics, such as age, gender, nationality, marital status, educational level, job role, experience, and income level, showed significant relationships with the study variables, creating a multifaceted representation of the nursing team. The correlations and descriptive statistics collectively contributed to discerning the nuanced relationships between demographic factors and job satisfaction, intention to stay, organizational commitment, and self-efficacy. This depth of analysis is reinforced by studies conducted by Marufu et al. [36], Park and Ko [37], and Woodward and Willgerodt [38], each shedding light on the intricate interplay between demographics and the four study variables in healthcare settings. Alshareef et al. [6], Warden et al. [20], and Woodward and Willgerodt [38], who delved into similar aspects in different healthcare settings, substantiate the findings of the present study. Their research indicated that variables such as age, job type, and tenure can impact the likelihood of nurses deciding to leave. The consistency across studies enhanced the robustness and reliability of the findings of the present study, extending beyond the confines of the specific hospital investigated. The findings provide valuable insights into the determinants of turnover among nurses at a hospital in Riyadh. By understanding the factors influencing nurse retention and their interrelationships, healthcare organizations can develop targeted interventions to enhance job satisfaction, organizational commitment, and self-efficacy, ultimately fostering a supportive work environment conducive to nurse retention. The findings underscore the importance of tailored retention strategies and the need to address diverse demographic factors to mitigate turnover challenges effectively. Overall, this comprehensive analysis offers valuable insights for healthcare organizations to foster an environment conducive to nursing retention and organizational success.

### Limitations and Strengths

One of the strengths of this study lies in its comprehensive investigation of the determinants of turnover among nurses, particularly within the specific context of a hospital in Riyadh. By employing a robust research design and utilizing validated measurement tools, this study ensured the reliability and validity of the data collected, thereby enhancing the credibility of the findings. Additionally, the inclusion of a diverse sample of nurses, encompassing various demographic characteristics and job roles, enriched the study’s insights and enhanced its generalizability to similar healthcare settings.

Furthermore, the meticulous data collection process and rigorous statistical analysis employed in this study contributed to the depth and rigor of the findings. By examining multiple variables such as job satisfaction, organizational commitment, intention to stay, and self-efficacy, this study offers a nuanced understanding of the factors influencing nurse turnover. The use of a correlation analysis provides insights into the relationships between these variables, shedding light on the complex interplay among different factors contributing to nurses’ decisions to stay or leave their positions. Overall, the methodological rigor and comprehensive approach of this study strengthened its contributions to the literature on nurse turnover and retention strategies.

While this study provides valuable insights into nurses’ perceptions of turnover determinants, certain limitations must be acknowledged. First, the findings are specific to the context of a hospital, and caution should be exercised when generalizing them to different healthcare settings. The cross-sectional nature of this research design captures a snapshot in time, thus limiting our ability to establish causation. Additionally, the reliance on self-reported data may introduce response biases because participants’ opinions may be understated or overstated. Despite these constraints, this study lays a foundation for future research to build upon and explore these dynamics more comprehensively.

## 5. Implications and Recommendations

The implications of this study extend to healthcare practitioners and organizational leaders. The positive correlations identified in this study underscore the interconnected nature of job satisfaction, intention to stay, organizational commitment, and self-efficacy, emphasizing the need for strategies that address these factors collectively. This, in turn, may mitigate turnover rates and contribute to a more stable and satisfied nursing workforce.

The implications of the study also span across multiple domains, including healthcare practice, policy, research, and education. In healthcare practice, understanding the factors influencing nurses’ intentions to stay or leave their positions is crucial for developing targeted interventions to enhance job satisfaction, organizational commitment, and self-efficacy among nursing staff. Practitioners can use these insights to create supportive work environments, provide opportunities for professional growth, and foster a culture of appreciation for nurses’ contributions, ultimately contributing to a more stable and satisfied nursing workforce. From a broader perspective, the findings have implications for healthcare policy, research, and education. Policymakers could leverage these insights to develop policies aimed at reducing turnover rates and promoting workforce stability, and researchers could delve deeper into the complex dynamics of nurse turnover and retention to identify additional influencing factors. In education, integrating contents related to workforce retention and job satisfaction into nursing curricula could better prepare future nurses to navigate the challenges of the healthcare workplace and advocate for supportive work environments. By addressing these implications collaboratively, stakeholders could work towards creating a more sustainable and fulfilling environment for nurses, thereby enhancing patient care outcomes and strengthening the healthcare system as a whole.

Several recommendations emerged for healthcare leaders and policymakers. First, initiatives aimed at improving job satisfaction, such as regular feedback mechanisms, professional development opportunities, and recognition programs, should be implemented. Fostering a positive work environment and supportive leadership could contribute to enhanced organizational commitment. Considering the demographic variations identified, tailored retention strategies for different age groups, job roles, and experience levels may prove beneficial. Regular assessments of work climate and employee satisfaction could inform ongoing interventions to address evolving concerns.

## 6. Conclusions

This study delved into and established the significant and positive relationship between job satisfaction, intention to stay, organizational commitment, self-efficacy, and demographic variables among clinical nurses. Despite encountering certain limitations inherent to research endeavors, the findings offer a robust foundation for further exploration in the field of nursing retention. The implications drawn from the study underscore the critical need for comprehensive strategies aimed at bolstering workplace satisfaction and fostering a supportive organizational culture conducive to nurse retention.

The recommendations stemming from the research emphasize the necessity for tailored interventions that account for the unique dynamics present within the nursing workforce. By addressing key factors, such as job satisfaction, organizational commitment, and demographic variables, healthcare institutions could strive towards mitigating turnover rates and cultivating a more stable and fulfilled nursing workforce. These recommendations advocate for a holistic approach to nurse retention, recognizing the multifaceted nature of the challenges faced within healthcare settings.

Ultimately, this study contributes significantly to the ongoing discourse surrounding nursing retention strategies and serves as a catalyst for future research endeavors in diverse healthcare contexts. By shedding light on the complex interplay of factors influencing nurse turnover, the findings offer valuable insights that could inform policy formulation, organizational practices, and educational initiatives aimed at enhancing nurse retention rates and fostering a sustainable healthcare workforce. An environment that supports the professional growth, well-being, and longevity of nurses in healthcare settings worldwide could be created through continued research and collaborative efforts.

## Figures and Tables

**Table 1 behavsci-14-01140-t001:** Demographic information (N = 227).

Variable	Category	Frequency	Percentage
Age (in years)	20–29	53	23.3%
	30–39	168	74.0%
	40–49	4	1.8%
	50–59	2	0.9%
Gender	Female	93	41.0%
	Male	134	59.0%
Nationality	Non-Saudi	65	28.6%
	Saudi	162	71.4%
Marital Status	Married	137	60.4%
	Single	90	39.6%
Education Level	Bachelor degree	57	25.1%
	Diploma	131	57.7%
	Doctorate	3	1.3%
	Postgraduate diploma	30	13.2%
	Master’s degree	6	2.6%
Job Role	Head nurse	1	0.4%
	General nurse	14	6.1%
	Nurse specialist	1	0.4%
	Nurse quality	1	0.4%
	LPN	173	76.2%
	Staff nurse	1	0.4%
	Supervisor	34	15.0%
Duration of Work Experience	1–5 years	67	29.5%
	11–15 years	17	7.5%
	6–10 years	62	27.3%
	Less than 1 year	75	33.0%
	More than 15 years	6	2.6%
Monthly Income	Less than 10,000 SAR	149	65.6%
	10,000–15,000 SAR	71	31.3%
	15,000–20,000 SAR	6	2.6
	More than 20,000 SAR	1	0.4%

**Table 2 behavsci-14-01140-t002:** Descriptive statistics of study variables (N = 227).

	k	α	M	(SD)	Range		Skewness		Kurtosis	
					Actual	Potential	Statistics	Std. Error	Statistics	Std. Error
Job Satisfaction	25	0.85	54.18	12.38	26–104	25–120	0.12	0.16	0.73	0.32
Intent to Stay	2	0.54	3.93	1.55	2–8	2–8	0.73	0.16	0.201	0.32
Organizational Commitment	15	0.76	31.42	7.24	15–60	15–60	–0.006	0.16	0.77	0.32
Self-Efficacy	10	0.83	23.17	7.24	11–40	10–40	0.69	0.16	0.35	0.32

**Table 3 behavsci-14-01140-t003:** Correlation among study variables (N = 227).

Study Variables	1	2	3	4
Job Satisfaction	-	0.362 **	0.657 **	0.578 **
Intent to Stay		-	0.486 **	0.489 **
Organizational Commitment			-	0.641 **
Self-Efficacy				-

** Significant at 0.01.

**Table 4 behavsci-14-01140-t004:** Correlation between nurses’ demographic characteristics and the four study variables (N = 227).

Demographic Characteristics	Job Satisfaction		Intention to Stay		Organizational Commitment		General Self-Efficacy	
	X^2^	*p*-Value	X^2^	*p*-Value	X^2^	*p*-Value	X^2^	*p*-Value
Age	221.97	0.001 ***	45.53	0.001 ***	127.32	0.029 *	219.48	0.001 ***
Gender	70.02	0.016 *	13.44	0.037 *	49.84	0.030 *	68.80	0.001 ***
Nationality	78.42	0.003 **	17.99	0.006 **	55.52	0.008 **	66.93	0.001 ***
Marital Status	67.01	0.029 *	18.29	0.006 **	56.53	0.007 **	55.50	0.004 **
Education Level	235.40	0.011 *	61.19	0.001 ***	213.61	0.001 ***	213.12	0.001 ***
Job Role	502.16	0.005 **	80.74	0.011 *	519.41	0.001 ***	503.41	0.001 ***
Duration of Work Experience	266.38	0.001 ***	60.70	0.001 ***	152.93	0.103	218.40	0.001 ***
Monthly Income	141.06	0.483	26.66	0.086	114.76	0.133	137.85	0.002 **

*** Significant at 0.001. ** Significant at 0.01. * Significant at 0.05.

## Data Availability

The data presented in this study are available upon request from the corresponding author.

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
