# Peer review of "Association of Job Satisfaction, Intention to Stay, Organizational Commitment, and General Self-Efficacy Among Clinical Nurses in Riyadh, Saudi Arabia"

_behavsci, 2024, doi:10.3390/bs14121140_

Round 1
Reviewer 1 Report
Comments and Suggestions for Authors
Thank you for the opportunity to review this manuscript. The study investigated the nurses’ perceptions of turnover determinants in a hospital in Riyadh and explored the relationships among job satisfaction, intention to stay, organizational commitment, and demographic variables. Generally, the study was well conducted, and the manuscript was written adequately. However, some issues need to be considered by the author to enhance the quality of the study:
1. Abstract: Aims: Can the authors please clarify the aim of the study? What does it mean by turnover determinants? Is it the same with determinants of nurse turnover? If yes, what are these determinants? Are the determinants the “job satisfaction, intention to stay, 15 organizational commitment, and demographic variables”? I think the aim can be stated clearly to avoid confusion.
2. Abstract: Methods: Please include the setting and data collection date.
3. Abstract: Methods: What tool was used in measuring turnover? The authors mentioned using the “Job Satisfaction Index, the Intent to Stay Scale, the Organizational Commitment Scale, and the General Self-Efficacy Scale.” Which among these tools was used to measure turnover? Is turnover synonymous with "intention to stay" in this study? Please clarify.
4. Abstract: Please review and ensure the consistency of the names of the study variables to avoid confusion.
5. Introduction: This section is well-written and provides a compelling rationale for conducting the study. This section also presents rigorous literature on international studies. However, the Introduction is so focused on “Turnover,” which is not a variable of the study. Also, in my humble opinion, what is lacking is a discussion of the research problem within the study’s context: in Saudi Arabia. I acknowledge that the Introduction presented a thorough discussion of the reason for conducting the study. Still, these reasons could become stronger if they are supported by literature from the country.
6. Purpose of the Study: Please ensure that the study's aims in the abstract and the text are consistent.
7. Sample: Please indicate the different parameters used in the sample size calculation. For example, what is the population size and the margin of error used in the calculation?
8. Data Collection Procedure: The authors mentioned that they collected data using Google Form, which was sent to the participants by the researcher. Can the authors clarify how they sent the online survey to the participants? Did they send it directly to the qualified participants via email or any social media? What are the potential implications of these methods for privacy and confidentiality?
9. Measurement: This is related to my comment on the study’s aim (the variable turnover). Based on this comment, the study measured the nurses' job satisfaction, intention to stay, organizational commitment, and general self-efficacy. I think these variables should be the focus of this study/ manuscript and not "turnover." I understand that the authors argue that these variables are "determinants of turnover"; however, the study did not measure turnover, which confuses the readers.
10. General Comment: I think what is confusing in this manuscript is the use of “turnover” and “determinants” in the objectives and title of the study. The title, objectives, and introduction are focused on “determinants of turnover,” but the study did not study turnover. It gives the impression that the study examined the determinants of turnover among nurses when the study really examined the nurses’ job satisfaction, intent to stay, organizational commitment, and self-efficacy, which are determinants of turnover according to the literature. This needs to be clarified throughout the manuscript to avoid confusing the readers. I suggest the authors focus on nurses’ job satisfaction, intent to stay, organizational commitment, self-efficacy, and the demographic variables that are associated with them rather than on turnover, which is not the focus of the study. I think the best way to move forward is to change the
a. Title to “Association of Job Satisfaction, Intent to Stay, Organizational Commitment, and General Self-efficacy among Clinical Nurses in Riyadh, Saudi Arabia”
b. Objectives to: This study examined the association of Job Satisfaction, Intent to Stay, Organizational Commitment, and General Self-efficacy among Clinical Nurses in Riyadh, Saudi Arabia. It also examined the sociodemographic variables associated with these variables.
c. The introduction needs to be reworked to focus on the study’s main variables: Job Satisfaction, Intent to Stay, Organizational Commitment, and General Self-efficacy. The authors may discuss how these variables influence nurses' turnover, but that should not be the focus of the introduction as that is not the direction of the study. The Introduction should provide a solid conceptual or theoretical framework or model of the relationships examined in the study.
d. Analysis: The authors should examine the sociodemographic variables associated with the nurses’ Job Satisfaction, Intent to Stay, Organizational Commitment, and General Self-efficacy.
e. Results and Discussion: The results on the sociodemographic variables associated with the nurses’ Job Satisfaction, Intent to Stay, Organizational Commitment, and General Self-efficacy should be reported and discussed.
f. Conclusion: The conclusion will be adjusted based on the objectives, analyses, and results revisions.
Comments on the Quality of English LanguageNone.
Author Response
Response to Reviewer 1 Comments
Thank you for the opportunity to review this manuscript. The study investigated the nurses’ perceptions of turnover determinants in a hospital in Riyadh and explored the relationships among job satisfaction, intention to stay, organizational commitment, and demographic variables. Generally, the study was well conducted, and the manuscript was written adequately. However, some issues need to be considered by the author to enhance the quality of the study:
RESPONSE: Dear Honorable Reviewer 1, we thank you very much for your positive comments and we highly appreciate them. We made our point-by-point response to your valuable comments below, and extensively revised our work based on them. We sincerely hope that our revisions will satisfy you and will help in convincing the Distinguished Editor to accept our paper for publication in Behavioral Sciences.
- Abstract: Aims: Can the authors please clarify the aim of the study? What does it mean by turnover determinants? Is it the same with determinants of nurse turnover? If yes, what are these determinants? Are the determinants the “job satisfaction, intention to stay, 15 organizational commitment, and demographic variables”? I think the aim can be stated clearly to avoid confusion.
RESPONSE: Thank you for this comment. We clearly stated the aim of our study as suggested (Refer to Lines 15-18), which is consistent with the aim and objectives in the main text (Refer to Lines 160-163).
- Abstract: Methods: Please include the setting and data collection date.
RESPONSE: Thank you for this comment. We provided the setting and data collection date. Please refer to Lines 20-21.
- Abstract: Methods: What tool was used in measuring turnover? The authors mentioned using the “Job Satisfaction Index, the Intent to Stay Scale, the Organizational Commitment Scale, and the General Self-Efficacy Scale.” Which among these tools was used to measure turnover? Is turnover synonymous with "intention to stay" in this study? Please clarify.
RESPONSE: Thank you for the opportunity to clarify that no tool was used to measure turnover. Hence, we corrected all phrases or statements in our study indicating or implying that turnover was measured, because we did not measure turnover at all in our study. Related to this, we also added the statement at the beginning of Discussion section, “However, it is critical to approach and interpret these findings with caution, as the investigation did not include the measurement of nurse turnover in this study.” Please refer to Lines 398-399.
- Abstract: Please review and ensure the consistency of the names of the study variables to avoid confusion.
RESPONSE: Thank you for this comment. This has been done, as suggested.
- Introduction: This section is well-written and provides a compelling rationale for conducting the study. This section also presents rigorous literature on international studies. However, the Introduction is so focused on “Turnover,” which is not a variable of the study. Also, in my humble opinion, what is lacking is a discussion of the research problem within the study’s context: in Saudi Arabia. I acknowledge that the Introduction presented a thorough discussion of the reason for conducting the study. Still, these reasons could become stronger if they are supported by literature from the country.
RESPONSE: Thank you for this comment. We revised this section extensively by adding systematic reviews, other international studies, and importantly, a discussion of the research problem in Saudi Arabian context. Please see Lines 78-91.
- Purpose of the Study: Please ensure that the study's aims in the abstract and the text are consistent.
RESPONSE: Thank you for this comment. The study’s aims both in Abstract and main text are now consistent.
- Sample: Please indicate the different parameters used in the sample size calculation. For example, what is the population size and the margin of error used in the calculation?
RESPONSE: Thank you for this comment. We have provided these parameters in Section 2.3 Study Population and Sampling (Refer to Lines 192-199).
- Data Collection Procedure: The authors mentioned that they collected data using Google Form, which was sent to the participants by the researcher. Can the authors clarify how they sent the online survey to the participants? Did they send it directly to the qualified participants via email or any social media? What are the potential implications of these methods for privacy and confidentiality?
RESPONSE: Thank you for this comment. We have provided additional information about this in Section 2.4 Procedure of Data Collection (Refer to Lines 219-228).
- Measurement: This is related to my comment on the study’s aim (the variable turnover). Based on this comment, the study measured the nurses' job satisfaction, intention to stay, organizational commitment, and general self-efficacy. I think these variables should be the focus of this study/ manuscript and not "turnover." I understand that the authors argue that these variables are "determinants of turnover"; however, the study did not measure turnover, which confuses the readers.
RESPONSE: Thank you so much for highlighting this and we sincerely apologize for the confusion. This has been corrected and clarified throughout the manuscript.
- General Comment: I think what is confusing in this manuscript is the use of “turnover” and “determinants” in the objectives and title of the study. The title, objectives, and introduction are focused on “determinants of turnover,” but the study did not study turnover. It gives the impression that the study examined the determinants of turnover among nurses when the study really examined the nurses’ job satisfaction, intent to stay, organizational commitment, and self-efficacy, which are determinants of turnover according to the literature. This needs to be clarified throughout the manuscript to avoid confusing the readers. I suggest the authors focus on nurses’ job satisfaction, intent to stay, organizational commitment, self-efficacy, and the demographic variables that are associated with them rather than on turnover, which is not the focus of the study.
RESPONSE: Thanks very much again for emphasizing this valuable aspect related to our study. We have redirected the focus of our manuscript on nurses’ job satisfaction, intent to stay, organizational commitment, self-efficacy, and the demographic variables that are associated with them.
I think the best way to move forward is to change the
- Title to “Association of Job Satisfaction, Intent to Stay, Organizational Commitment, and General Self-efficacy among Clinical Nurses in Riyadh, Saudi Arabia”
RESPONSE: RESPONSE: Thank you very much and we revised the title based on this.
- Objectives to: This study examined the association of Job Satisfaction, Intent to Stay, Organizational Commitment, and General Self-efficacy among Clinical Nurses in Riyadh, Saudi Arabia. It also examined the sociodemographic variables associated with these variables.
RESPONSE: Thank you very much for this valuable suggestion. We have revised the objectives based on this. Please see Lines 160-163.
- The introduction needs to be reworked to focus on the study’s main variables: Job Satisfaction, Intent to Stay, Organizational Commitment, and General Self-efficacy. The authors may discuss how these variables influence nurses' turnover, but that should not be the focus of the introduction as that is not the direction of the study. The Introduction should provide a solid conceptual or theoretical framework or model of the relationships examined in the study.
RESPONSE: Thank you very much for this comment. We have reworked the Introduction section to focus on the study’s main variables: job satisfaction, intent to stay, organizational commitment, and general self-efficacy. We have done extensive revision on the section (Refer to Lines 58-142).
- Analysis: The authors should examine the sociodemographic variables associated with the nurses’ Job Satisfaction, Intent to Stay, Organizational Commitment, and General Self-efficacy.
RESPONSE: Thank you for this comment. We have added this as Table 4, as suggested. Please see Lines 388-389.
- Results and Discussion: The results on the sociodemographic variables associated with the nurses’ Job Satisfaction, Intent to Stay, Organizational Commitment, and General Self-efficacy should be reported and discussed.
RESPONSE: Thank you for this comment. This has been reported in Lines 378-386, and discussed in Lines 469-492.
- Conclusion: The conclusion will be adjusted based on the objectives, analyses, and results revisions.
RESPONSE: Thank you so much once again for your valuable comments and we have adjusted the conclusion based on the objectives, analyses, and results revisions. Please refer to Lines 578-580.

Reviewer 2 Report
Comments and Suggestions for Authors
OVERALL COMMENTS: I am thankful being invited and given the opportunity to review this interesting topic that would add scientific value to enhancing nursing retention because of the ongoing shortage of nursing staff globally and particularly in the setting of the current study. Presenting the associations of four variables as determinants of nursing turnover including job satisfaction, intent to stay, organizational commitment, and general self-efficacy adds to the scientific knowledge of nursing. The presentation and discussion of the results as well as the implications and recommendations are extensive and well supported with literature. I have some suggestions for the betterment of the current version of the paper so that it would benefit eventual publication. Appropriate consideration in the revision is expected from the authors.
SPECIFIC/DETAILED COMMENTS:
ABSTRACT
Line 24 – How about intent to stay and self-efficacy? I think they are missing here.
INTRODUCTION
Needs to establish that the following variables are the determinants of nurse turnover, including job satisfaction, intent to stay, organizational commitment, and general self-efficacy.
Lines 53-59 (entire paragraph) and Lines 60-65 (entire paragraph) – can be merged.
Line 66 – consider correcting the phrase ‘this was study conducted…’ to ‘this study was conducted…’ and adding compelling justification locally in Saudi Arabia why the study is needed. The other reasons/justifications are well presented and substantiated.
MATERIALS AND METHODS
Lines 123-126 – would benefit adding the STROBE guidelines and suggesting to submit the checklist as supplementary file. You may find it here: https://www.strobe-statement.org/checklists/
Lines 128-138 – would be enhanced by adding the source.
Lines 141-142 – ‘practical nurses…neuro)’ consider deleting here because it is repeated in Lines 151-153.
Lines 175-177 – would benefit from adding the validity and reliability of the tool.
Lines 188-193 – suggesting that it would be incorporated in the ‘Measurements’ section.
Lines 194-196 - suggesting transfer to ‘Data Analysis’ section.
Line 212 – would the authors consider this ‘kept private’ as ‘kept anonymous’?
Lines 218-219 – would the authors consider revising this for clarity?
RESULTS
Line 222 – why only 99%?
Lines 221-238 – consider revising this by presenting the majority.
Lines 266-284 – suggesting that the authors present the actual p-values for transparency.
Lines 290-294 – would consider transferring it as first part of the ‘Discussion’ section
Lines 295 -297 – would consider it as one of the implications of the study
DISCUSSION
Overall, this section would benefit better organization by following sequence of argument based on the results per variables.
Lines 346-351 and 358-366 – I think the authors missed to include the data in the ‘Results’ section? I suggest adding these data since the authors already included a discussion about these results.
IMPLICATIONS AND RECOMMENDATIONS
Consider merging Lines 422-429 and Lines 430-439.
CONCLUSIONS
Consider deleting ‘within Prince Mohammed Bin Abdulaziz Hospital in Riyadh’ in Line 452.
OTHERS
To protect the identity of the hospital setting, would the authors consider confidentiality of the hospital by referring ‘Prince Mohammed Bin Abdulaziz Hospital in Riyadh’ as ‘hospital in Riyadh’ in the main text?
Author Response
Response to Reviewer 2 Comments
OVERALL COMMENTS: I am thankful being invited and given the opportunity to review this interesting topic that would add scientific value to enhancing nursing retention because of the ongoing shortage of nursing staff globally and particularly in the setting of the current study. Presenting the associations of four variables as determinants of nursing turnover including job satisfaction, intent to stay, organizational commitment, and general self-efficacy adds to the scientific knowledge of nursing. The presentation and discussion of the results as well as the implications and recommendations are extensive and well supported with literature. I have some suggestions for the betterment of the current version of the paper so that it would benefit eventual publication. Appropriate consideration in the revision is expected from the authors.
RESPONSE: Dear Esteemed Reviewer 2, we thank you so much for your positive comments and we highly appreciate them. We made our point-by-point response to your valuable comments below, and extensively revised our work based on them. Also, please know that some of your valuable comments were addressed when we revised our work based on the comments of the Honorable Reviewer 1. We sincerely hope that our revisions will satisfy you and will help in convincing the Distinguished Editor to accept our paper for publication in Behavioral Sciences.
SPECIFIC/DETAILED COMMENTS:
ABSTRACT
Line 24 – How about intent to stay and self-efficacy? I think they are missing here.
RESPONSE: Thank you for highlighting this and we sincerely apologize for missing them. We have added and corrected this part of the Abstract in Lines 27-35.
INTRODUCTION
Needs to establish that the following variables are the determinants of nurse turnover, including job satisfaction, intent to stay, organizational commitment, and general self-efficacy.
RESPONSE: Thank you very much for this comment. We have extensively revised the Introduction based on this. Please refer to Lines 58-142.
Correspondingly, the citations and order of the references have been updated based on the journal guidelines.
Lines 53-59 (entire paragraph) and Lines 60-65 (entire paragraph) – can be merged.
RESPONSE: Thank you for this comment. This was addressed during the extensive revision made on the Introduction section, based on the comment of the Honorable Reviewer 1.
Line 66 – consider correcting the phrase ‘this was study conducted…’ to ‘this study was conducted…’ and adding compelling justification locally in Saudi Arabia why the study is needed. The other reasons/justifications are well presented and substantiated.
RESPONSE: Thank you so much for this important comment. This has been addressed in Lines 78-92.
MATERIALS AND METHODS
Lines 123-126 – would benefit adding the STROBE guidelines and suggesting to submit the checklist as supplementary file. You may find it here: https://www.strobe-statement.org/checklists/
RESPONSE: Thank you for this comment. This has been addressed in Lines 175-177 and the STROBE checklist for our study has been uploaded to the MPDI system as Supplementary File 1. Please note that the page number/s indicated in the checklist is based on the revised version file with tracked changes. The page number/s will be updated base on the author proof file, if our paper will be hopefully accepted for publication.
Lines 128-138 – would be enhanced by adding the source.
RESPONSE: Thank you for this comment. The source has been added in Line/s 182.
Lines 141-142 – ‘practical nurses…neuro)’ consider deleting here because it is repeated in Lines 151-153.
RESPONSE: Thank you for this comment. This has been deleted, as suggested. Please refer to Lines 193-194.
Lines 175-177 – would benefit from adding the validity and reliability of the tool.
RESPONSE: Thank you for this comment. This information has been added. Please see Lines 244-245.
Lines 188-193 – suggesting that it would be incorporated in the ‘Measurements’ section.
RESPONSE: Thank you for this comment. This has been transferred to Measurement section in Lines 255-259.
Lines 194-196 - suggesting transfer to ‘Data Analysis’ section.
RESPONSE: Thank you for this comment. This has been deleted because similar information was found in Lines 272-277.
Line 212 – would the authors consider this ‘kept private’ as ‘kept anonymous’?
RESPONSE: Thank you for this comment. We changed it to ‘completely kept anonymous,’ in Lines 284-285.
Lines 218-219 – would the authors consider revising this for clarity?
RESPONSE: Thank you for this comment. This has been revised, as suggested, in Lines 290-291.
RESULTS
Line 222 – why only 99%?
RESPONSE: Thank you for this comment. Justification has been added in Lines
Lines 221-238 – consider revising this by presenting the majority.
RESPONSE: Thank you for this comment. This has been revised, as suggested, in Lines 295-302.
Lines 266-284 – suggesting that the authors present the actual p-values for transparency.
RESPONSE: Thank you for this comment. Actual p-values have been presented, as suggested, in Lines 353-368.
Lines 290-294 – would consider transferring it as first part of the ‘Discussion’ section
RESPONSE: Thank you for this comment. This has been addressed during the revision made, based on the comment of the Honorable Reviewer 1.
Lines 295 -297 – would consider it as one of the implications of the study
RESPONSE: Thank you for this comment. This has been incorporated in the Implications and Recommendations section.
DISCUSSION
Overall, this section would benefit better organization by following sequence of argument based on the results per variables.
RESPONSE: Thank you very much for this comment. The flow of discussion has been revised, as suggested.
Lines 346-351 and 358-366 – I think the authors missed to include the data in the ‘Results’ section? I suggest adding these data since the authors already included a discussion about these results.
RESPONSE: Thank you for this comment and at the same time, we sincerely apologize because we decided not to include this finding (comparison/difference) and realized to keep the focus of our study’s objectives. So, we deleted this part.
IMPLICATIONS AND RECOMMENDATIONS
Consider merging Lines 422-429 and Lines 430-439.
RESPONSE: Thank you for this comment and we merged these parts.
CONCLUSIONS
Consider deleting ‘within Prince Mohammed Bin Abdulaziz Hospital in Riyadh’ in Line 452.
RESPONSE: Thank you so much for this comment. We deleted name of the hospital, ‘Prince Mohammed Bin Abdulaziz Hospital,’ in this part, as suggested.
OTHERS
To protect the identity of the hospital setting, would the authors consider confidentiality of the hospital by referring ‘Prince Mohammed Bin Abdulaziz Hospital in Riyadh’ as ‘hospital in Riyadh’ in the main text?
RESPONSE: Thank you so much for this comment. We changed the exact name of the hospital, ‘Prince Mohammed Bin Abdulaziz Hospital’ and referred it as ‘hospital’ throughout the main text.
OTHER RESPONSE: Since there were added studies to further support our argument, discussion and presentation, we updated the citations and order of the references, and adhered to the journal guidelines.
Round 2
Reviewer 1 Report
Comments and Suggestions for Authors
The authors considered my comments and comprehensively revised the manuscript. It is now ready for publication. Congratulations!